# Synthesis, Electronic Structure, and Electrochemical Properties of the Cubic Mg_2_MnO_4_ Spinel with Porous-Spongy Structure

**DOI:** 10.3390/nano11051122

**Published:** 2021-04-27

**Authors:** Zhenyan Wang, He Zhu, Li Ai, Jimin Ding, Pengfei Zhu, Ziqing Li, Bo Li, Hechun Jiang, Fapeng Yu, Xiulan Duan, Huaidong Jiang

**Affiliations:** 1State Key Laboratory of Crystal Materials, Institute of Crystal Materials, Shandong University, Jinan 250100, China; 18839199546@163.com (Z.W.); zhuhe@mail.sdu.edu.cn (H.Z.); 201912593@mail.sdu.edu.cn (L.A.); jiminding@mail.sdu.edu.cn (J.D.); joseph.zhu.nt@gmail.com (P.Z.); lzq@fudan.edu.cn (Z.L.); boli@sdu.edu.cn (B.L.); jianghechun001@126.com (H.J.); fapengyu@sdu.edu.cn (F.Y.); jianghd@shanghaitech.edu.cn (H.J.); 2School of Physical Science and Technology, ShanghaiTech University, Shanghai 201210, China

**Keywords:** cubic spinel, Mg_2_MnO_4_, inversion degree, Li-ion batteries, anode materials

## Abstract

Mg_2_MnO_4_ nanoparticles with cubic spinel structure were synthesized by the sol-gel method using polyvinyl alcohol (PVA) as a chelating agent. X-ray powder diffraction, infrared spectrum (IR), scanning electron microscope (SEM), and transmission electron microscope (TEM) were used to characterize the crystalline phase and particle size of as-synthesized nanoparticles. The electronic structure of Mg_2_MnO_4_ spinel was studied by X-ray photoelectron spectroscopy (XPS). The results showed that pure cubic Mg_2_MnO_4_ spinel nanoparticles were obtained when the annealing temperature was 500–700 °C. The samples had a porous-spongy structure assembled by nanoparticles. XPS studies indicated that Mg_2_MnO_4_ nanoparticles were mixed spinel structures and the degree of cation inversion decreased with increasing annealing temperature. Furthermore, the performance of Mg_2_MnO_4_ as lithium anode material was studied. The results showed that Mg_2_MnO_4_ samples had good cycle stability except for the slight decay in the capacity at 50 cycles. The coulombic efficiency (ratio of discharge and charge capacity) in most cycles was near 100%. The sample annealed at 600 °C exhibited good electrochemical properties, the first discharge capacity was 771.5 mAh/g, and the capacity remained 340 mAh/g after 100 cycles. The effect of calcination temperature on the charge–discharge performance of the samples was studied and discussed.

## 1. Introduction

Spinel oxides with the general formula of AB_2_O_4_ (A and B are the metal cations) are important semiconductor materials and have been widely used in the fields of electronics, ceramics, communication, optics, magnetism, and catalysis because of the advantages of corrosion resistance, high hardness, good thermal stability, high melting point [1,2,3,4,5]. In the structure of AB_2_O_4_ spinel, each cell contains eight AB_2_O_4_ molecules, that is, thirty-two O^2−^ ions, sixteen B^3+^ ions, and eight A^2+^ ions. The spinel structure can be considered as oxygen ions arranged in a cubic close-packed structure, forming tetrahedral and octahedral vacancies. The A^2+^ and B^3+^ cations are distributed in these two sites [6]. Considering the cation distribution, the spinel oxide can also be represented by the general formula (A_1-x_B_x_) [A_x_B_2-x_] O_4_, where the parentheses ( ) and brackets [ ] represent the tetrahedral and octahedral sites, respectively, and x denotes the inversion parameter. Usually, there are three types of spinel: (1) normal spinel (x = 0), where A ions occupy the tetrahedral sites and B ions occupy the octahedral sites; (2) inverse spinel (x = 1), where the A and one-half of the B cations are in the octahedral sites and the rest of B ions enter the tetrahedral sites; (3) mixed spinel (0 < x < 1), where both the A and B ions are randomly situated at both the sites. The distribution of cations in the spinel is affected by many factors, such as the type and nature of cations, synthesis temperature of materials, particle size, and impurity content [7,8]. The cation distribution of spinel materials has great effects on their properties, such as magnetism, catalytic activity, electrochemical activity. The relationship between the structure and properties can be better understood by studying the distribution of metal cations in the spinel [8,9,10,11].

Among many spinel materials, the spinels containing manganese have attracted much attention and are widely used in the field of secondary batteries and photoelectric catalysis because manganese has a plurality of valence states and is low in price, non-toxic, and environment-friendly. Studies showed that spinels containing a higher content of tetravalent manganese were beneficial to ion migration and could be used as excellent materials for electrochemical cells or rechargeable batteries [3,12]. The Mg_2_MnO_4_ spinel has a cubic structure, and it has proven to be a good oxygen evolution photoanode [13,14]. Mg_2_MnO_4_ is considered an inverse spinel structure; half of the magnesium ions are located in the tetrahedral site, and half of the magnesium ions and manganese ions are located in the octahedral site together [15]. The structure diagram is shown in Figure 1. Actually, the distribution of magnesium and manganese ions in Mg_2_MnO_4_ is influenced by many factors, such as heat treatment temperature and time, and ion doping.

Mg_2_MnO_4_ has been previously prepared by solid-state reactions, coprecipitation method, and microemulsion method [16,17]. The pure phase of Mg_2_MnO_4_ cannot be obtained by the former two methods. Although the material prepared by the microemulsion method is relatively pure Mg_2_MnO_4_, the synthesis process is complicated and difficult to control. In this study, pure Mg_2_MnO_4_ spinel nanoparticles with a cubic structure were synthesized by a simple and easy to operate sol-gel method using polyvinyl alcohol (PVA) as the chelating agent and characterized by XRD, SEM, TEM, and IR techniques. The electronic structure and cation distribution of the Mn and Mg in the Mg_2_MnO_4_ spinel were studied by X-ray photoelectron spectroscopy. The as-synthesized Mg_2_MnO_4_ nanoparticles were used for the first time as anode material for Li-ion batteries, and the electrochemical properties were determined.

## 2. Materials and Methods

### 2.1. Synthesis

Mg_2_MnO_4_ nanoparticles with the structure of a cubic spinel were synthesized by the sol-gel method using polyvinyl alcohol (PVA, Sinopharm Chemical Reagent Co., Ltd., Shanghai, China) as the chelating agent. The detailed experimental process was as follows: magnesium nitrate (≥99%, Sinopharm Chemical Reagent Co., Ltd., Shanghai, China) and manganese acetate (≥99%, Sinopharm Chemical Reagent Co., Ltd., Shanghai, China) were firstly dissolved in deionized water to form a transparent solution, polyvinyl alcohol was added to the solution under stirring. The molar ratio of PVA to metal ions was 2:1. After mixing and thoroughly stirring, the solution was heated on the magnetic heating agitator until viscous gels were formed. The gels were then dried at 110 °C in an oven and calcined at different temperatures (500–800 °C) for 5 h to obtain the target products.

### 2.2. Characterization

The X-ray diffraction (XRD) data of the Mg_2_MnO_4_ nanoparticles were collected by a D8-Advance type multi-functional powder diffractometer manufactured by Bruker, Germany. Cu target K radiation and carbon monochromator were used in the measurement. The micromorphology of the samples was observed by a ZEISS scanning electron microscope made in Germany. The micrographs of the powders were observed by a high-resolution transmission electron microscope using a JEM-2100F type field produced by Japan. The infrared absorption spectra of the samples were obtained by the NEXUS 670-type Fourier transform Infrared-Raman spectrometer manufactured by Thermo Nicolet. N_2_ adsorption and desorption tests were mainly used to characterize the specific surface area of the sample. In this experiment, a JW-BK112T automatic gas adsorption analyzer (Beijing Jingwei Gaobo Science and Technology Co., Ltd., Beijing, China) was used for testing.

X-ray photoelectron spectra (XPS) were measured by a Thermofisher ESCALAB 250 (Waltham, MA, USA) X-ray photoelectron spectrometer with the monochromatic Al Kα X-ray source in an ultrahigh vacuum (≤10^−7^ Pa). The spot size was 500 μm. The neutralizing gun was opened during the test to neutralize the excess positive charge on the sample surface, and the binding energies of all the spectra were calibrated by using the C 1 s peak (284.6 eV) of carbon impurities as a reference. The spectra were deconvoluted after background subtraction, using a mixed Gaussian–Lorentzian function. Fractional atomic concentrations of the elements were calculated using empirically derived atomic sensitivity factors.

### 2.3. Electrochemical Characterization

Electrochemical measurements of the cubic phase Mg_2_MnO_4_ nanoparticles were performed by making coin cells. The working electrode was made by using as-synthesized Mg_2_MnO_4_ nanoparticles as the active material. Firstly, Mg_2_MnO_4_, acetylene black, and polyvinylidene difluoride were mixed at a mass ratio of 8:1:1 and then dissolved in n-methyl pyrrolidone (NMP). The above slurry was evenly coated on the hydrophilic carbon cloth and dried in a vacuum drying oven at 120 °C for 12 h. The lithium plate was used as the opposite electrode; Celgard 2300 and lithium hexafluorophosphate solutions were used as the separator and electrolyte, respectively. All the above operations were carried out in a glove box filled with argon gas.

The performance of the cells was evaluated in the voltage range of 0–3 V on a LAND battery test system (CT2001A, Wuhan Lambo Test Equipment U Co., Ltd., Wuhan, China), and the charge–discharge current was kept at 0.5 mA. An electrochemical workstation (CHI660E, Shanghai Chenhua Instrument Co., Ltd., Shanghai, China) was used to test the cyclic voltammetry (CV) and electrochemical impedance spectroscopy (EIS). The CV test voltage range was 0.1–3 V, and the scanning speed was 0.3 mV/s. The EIS was recorded in the frequency range from 1 MHz to 10 mHz with an amplitude of 10 mV.

## 3. Results and Discussion

### 3.1. Preparation of Mg_2_MnO_4_ Nanoparticles

X-ray powder diffraction was used to identify the phase of the as-synthesized samples. Figure 2 shows the XRD patterns of all the powder samples. As shown in Figure 2, several obvious diffraction peaks appeared in the XRD pattern of the sample annealed at 500 °C, indicating the formation of nanocrystals. The peak intensity increased gradually with the increase in annealing temperature, indicating that the crystallization degree of the sample increased. All the peaks indexed as (111), (220), (311), (400), (511), (440), and (531) crystal planes were assigned to the cubic Mg_2_MnO_4_ phase (ICDD PDF#19-0773). When the annealing temperature was up to 800 °C, a new weak impurity peak of Mg_6_MnO_8_ (No.19-0766) occurred. The XRD result showed that the pure cubic Mg_2_MnO_4_ spinel could be prepared by the sol-gel method when the annealing temperature was 500–700 °C. In addition, based on the XRD data, the cell parameters of the samples were estimated according to the formula: $a=d\sqrt{h^{2}+k^{2}+l^{2}}$, where h, k, and l are the Miller indices, d is the interplanar spacing, which is obtained by the Bragg equation (2dsinθ=λ). The parameters were 8.285 (±0.001) Å, 8.298 (±0.001) Å, 8.330 (±0.001) Å, and 8.334 (±0.001) Å for the samples annealed at 500–800 °C.

The morphology and particle size of the Mg_2_MnO_4_ samples were observed by SEM and TEM, shown in Figure 3 and Figure 4, respectively. From the inset of Figure 3a, it can be seen that the sample annealed at 500 °C had a porous-spongy structure, the samples annealed at 600–800 °C had a similar porous-spongey structure (Appendix A). When the magnification was increased, it was found that the structure was assembled by nanoparticles (Figure 3a), and all the annealed samples showed a similar microstructure (Figure 3b–d). It can be seen from Figure 4a,b that the particle size of the samples annealed at 500–600 °C was in the range of 30–50 nm. When the annealing temperature was increased to 700–800 °C, the particles of the sample grew rapidly, and the size was up to 100–200 nm. The high-resolution (HR) TEM image was taken for the particles shown in Figure 4e. The lattice spacing of approximately 0.48 nm corresponded to the d spacing of the (111) crystal plane of the cubic Mg_2_MnO_4_. Figure 5 shows the N_2_ adsorption/desorption isotherms at different annealing temperatures. The specific surface areas of the samples were calculated by the BET method to be 61.04 cm^3^/g, 34.95 cm^3^/g, 11.33 cm^3^/g, and 8.59 cm^3^/g for the temperatures 500–800 °C, indicating that with the increase in the annealing temperature, the specific surface area decreased because of the increasing particle size, which was confirmed by the TEM results.

The IR spectra of the sintered samples are presented in Figure 6. From the spectrum of the precursor (Figure 6), we could see that there were several intense absorption peaks: the absorption peak at 1450–1700 cm^−1^ was caused by the deformation and vibration of water, the wide absorption peak at 3400 cm^−1^ was due to the stretching vibration of lattice -OH, and the asymmetric stretching of CO_2_ appeared at 2400 cm^−1^ [18]. When the sample was annealed at 500 °C, an obvious absorption peak at 695 cm^−1^ appeared in the IR spectra, and the new peak was the characteristic absorption peak of Mg_2_MnO_4_. When the heat treatment temperature of the sample was increased to 600 °C, the peak at 695 cm^−1^ was enhanced, indicating the crystallization was strengthened. The result was consistent with the XRD analysis.

### 3.2. XPS Studies

The survey XPS spectra of Mg_2_MnO_4_ samples annealed under different temperatures were determined and shown in Figure 7a. The results showed that no other elements were detected except for the original components and contaminated carbon. The C 1 s peak of the carbon contamination at 284.6 eV was used as reference (Figure 7b). The O 1 s spectra (Figure 7c) could be deconvoluted into two peaks. The peak located at the high-binding energy side (530.6 eV) was mainly from the adsorbed oxygen, while the peak with the low-binding energy (529.6) was allocated to the metal-oxygen bond corresponding to the lattice oxygen [19]. In order to analyze the chemical environments and valence states of Mn and Mg ions, Mn 2p_3/2_ and Mg 1 s high-resolution photoelectron spectra were measured, and the results are shown in Figure 7d,e, respectively.

XPS spectra of Mn 2p_3/2_ were wide and asymmetric, indicating that Mn ions occupied more than one coordination environment or had different valence states in the samples. In order to analyze the distribution of Mn ion in the sample, XPS spectra of Mn 2p_3/2_ were deconvolved; the analysis results are shown in Figure 7d. We fixed the binding energy difference between the two adjacent peaks of Mn 2p_3/2_ to 1.5 eV to obtain accurate values. After deconvolution, the Mn 2p_3/2_ orbital consisted of three peaks. The data of the binding energy, full width at half maximum, and relative content of Mn 2p_3/2_ are listed in Table 1. It is generally known that Mn has a plurality of valence states. It was reported that the binding energy values of Mn^2+^, Mn^3+^, and Mn^4+^ were in the ranges of 640.6–641.2 eV, 642.0–642.5 eV, and 643.5–644.2 eV, respectively [20,21,22]. Therefore, the three peaks of Mn 2p_3/2_ orbital at about 640.8 eV, 642.3 eV, and 643.8 eV corresponded to Mn^2+^, Mn^3+^, and Mn^4+^, respectively [23,24]. Similarly, the XPS spectra of Mg 1 s were analyzed. After deconvolution, the analysis results are shown in Figure 7e. Mg 1 s orbital was divided into two peaks, indicating that Mg^2+^ ions occupy two different sites in Mg_2_MnO_4_ spinel. The binding energy data are listed in Table 2. The peaks at 1302.6 eV and 1302.2 eV were attributed to Mg ions at octahedral and tetrahedral positions, respectively [25,26,27].

The change of cation distribution in the microstructure of the material led to lattice distortion, which had a macro effect on the properties of the material. We analyzed the cationic space occupation and the reversion degree of the samples at different sintering temperatures. The theoretical structure of the cubic Mg_2_MnO_4_ could be described as (Mg^2+^) [Mn^4+^_1/2_Mg^2+^_1/2_]_2_O_4_. However, due to the influence of the heat treatment temperature, synthesis method, and other experimental conditions, manganese ions were many more than one valence state and chemical environments in manganese-containing spinels [24,28]. According to the report, the absolute Octahedral Site Preference Energy (OSPE) value of Mn^3+^ (95.4 kJ mol^−1^) was higher than that of Mn^2+^(0 kJ mol^−1^), indicating that Mn^3+^ and Mn^4+^ tended to occupy the octahedral site and Mn^2+^ tended to occupy the tetrahedral site [1,29]. Considering the distribution of Mn ions with different valence states, the general formula of Mg_2_MnO_4_ could be described as (Mg^2+^_x_Mn^2+^_1-x_) [Mn^3+/4+^_x/2_Mg^2+^_2-x/2_] O_4_. It can be seen from Table 1 that the percent of Mn^4+^ ions was 28.85% for the 500 °C heated sample and decreased with the increase in heat treatment temperature. The sum of Mn^3+^ and Mn^4+^ at the octahedral sites decreased from 87% to 80% when the sintering temperature increased from 500 °C to 800 °C. According to the distribution of manganese ions in the sample, the inversion degree calculated is listed in Table 1. It could be found that with the increase in heat treatment temperature of the sample, the inversion degree dropped from 0.87 to 0.80. Similarly, the inversion degree calculated according to the distribution of magnesium ions in the sample is listed in Table 2. By comparing the inversion parameters calculated at different temperatures, it could be found that the inversion parameters calculated for Mn and Mg ions were consistent, and the value decreased with the increase in the temperature. The result indicated that as the temperature increased, more Mn^2+^ ions emerged and occupied the tetrahedral sites in the samples. Meanwhile, the fraction of Mg^2+^ ions in the tetrahedral sites was reduced. As a result, we concluded that the sample was a mixed spinel and the degree of cationic disorder decreased as the temperature increased.

### 3.3. Electrochemical Properties

The curves of the charge–discharge performance of the Mg_2_MnO_4_ samples annealed at different temperatures are shown in Figure 8. All the samples showed good cycle stability, and the coulombic efficiency was near 100%. The first discharge capacities were 537.0, 771.5, 531.7, and 522 mAh/g for the samples annealed at 500–800 °C. The capacity was higher than the commercial graphite anode materials, for which the theoretical specific capacity was 372 mAh/g [30]. It can be seen from Figure 8 that the capacities increased with the temperature increasing from 500 °C to 600 °C, and then decreased when the temperature was higher than 700 °C. This change could be explained in combination with the particle size and microstructure of the samples. From the above analysis, we knew that the samples annealed at 500–600 °C had a small crystallite size (30–50 nm), which was helpful for the electrochemical properties. From the view of microstructure, the sample annealed at 600 °C had a relatively small inversion degree when compared with the 500-heated sample. The previous studies indicated that the spinel materials with less inversion degree showed better electrochemical performance [31,32]. Therefore, the 600-heated sample displayed the higher capacity. However, the crystal particle of the sample grew rapidly when annealed at 700–800 °C, and the size was more than 100 nm; thus, the specific surface area decreased greatly. Although the inversion degree of the samples decreased with increasing annealing temperature, the large crystallite particle was not conducive to the migration of lithium ions in the material, so the discharge capacity of the sample annealed at these two temperatures decreased. After 100 cycles, the sample annealed at 600 °C remained the largest capacity of 340.0 mAh/g, and all the samples showed almost no decay in the capacities; the cycle stability of Mg_2_MnO_4_ was better than that of the MgCo_2_O_4_ spinel [33]. As a result, the sample annealed at 600 °C as lithium battery anode material had a good electrochemical performance and showed a highest charge–discharge capacity.

The detailed electrochemical properties of the 600-annealed sample were measured and shown in Figure 9. Figure 9a shows the capacity retention of Mg_2_MnO_4_ electrode materials. It could be seen that the discharge capacities were 343, 290, 166, 104, and 80 mAh/g when the current densities were 50, 100, 400, 800, and 1000 mA/g, respectively. When the current density returned to 100 mAh/g, the discharge capacity restored to 290 mAh/g, indicating that Mg_2_MnO_4_ had a good capacity recovery. From the cycle volt–ampere curve of the Mg_2_MnO_4_ sample shown in Figure 9b, it could be seen that the pairs of redox peaks of Mg_2_MnO_4_ appeared in the first three cycles. The position of the oxidation peak in the first cycle was about 0.4 V, and the position increased to 0.7 V for the second and third cycles. The position of the reduction peak in the first three cycles was basically kept the same value at 0.63 V. The results indicated that Mg_2_MnO_4_ as a lithium negative electrode had good electrochemical reversibility.

Figure 9c shows the voltage-capacity distribution curves of the sample at cycles 1, 2, 3, 5, 50, 100, and 150. We could see that in the 2, 3, 5, 100, and 150 cycles, the sample exhibited almost no capacity attenuation and good cycle stability. However, at the 50th cycle, there was a capacity fading, but this phenomenon returned to its previous state in later cycles. By comparing the voltage–capacity distribution curves at 500 °C, 700 °C, and 800 °C (Appendix A), it was found that the samples at different annealing temperatures all experienced a certain degree of capacity attenuation at 50 cycles. In order to explore the reason for this phenomenon, we conducted an electrochemical impedance analysis, morphology, and phase analysis on the samples at 50 cycles. Figure 9d is the EIS diagram of the material in a fresh cell and after 50 cycles. Figure 9d shows the electrochemical impedance spectra of Mg_2_MnO_4_ electrode material in a fresh cell and after 50 cycles. Firstly, in the equivalent circuit, R_Ω_ represented the internal resistance of the battery, mainly including electrolyte resistance (R_e_), surface film resistance (R_sf_), etc., C_d_ represented the double-layer capacitance, and Z_f_ represented the Faraday impedance. The Faraday impedance could be further divided into two parts: the charge transfer resistance (R_ct_) and the Warburg impedance (Z_W_) [34,35]. Nyquist plots showed that the Mg_2_MnO_4_ electrode presented a semicircular arc and an inclined line distributed from high frequency to low frequency, indicating that, in the high-frequency region, the electrode process was controlled by charge transfer, while in the low-frequency region, the electrode process was dominated by mass transfer [36]. The diameter of the semicircular arc in the high-frequency region of the Nyquist diagram corresponded to the internal resistance of the battery. The impedance for the fresh cell of Mg_2_MnO_4_ was estimated to be 12 ohm; this indicated that the material had a high electronic conductivity. After 50 cycles, the impedance of the electrode was changed to 24 ohm, indicating that the conductivity of the material decreased, and the resistance increased. It was speculated that the reason for this phenomenon was the change of the morphology and structure of the material. In order to verify this assumption, the electrode material after 50 cycles was tested by SEM and XRD, and the results are shown in Figure 9e,f. According to the SEM figure (Figure 9e), the morphology of the material collapsed. XRD results showed that the phase of the material changed after 50 cycles, part of the cubic phase Mg_2_MnO_4_ changed into Mg_0.9_Mn_0.1_O and MgO, some manganese ions combined with lithium ions in the electrolyte to form Li_0.4_Mn_0.6_O, the diffraction peaks around 2θ = 22° and 25 ° belonged to the carbon cloth collector. Presumably, this change was reversible because the capacity returned to its previous state in the later cycles. The voltage–capacity curves of the samples at 500 °C, 700 °C, and 800 °C showed similar capacity attenuation at about 50 cycles (Appendix A). By comparing the changes of EIS, SEM, and XRD of the fresh cell and after 50 cycles (Appendix A), all the samples showed a similar process of change.

## 4. Conclusions

Using PVA as the chelating agent, and Mg_2_MnO_4_ with cubic spinel structure was successfully prepared by the sol-gel method. XRD, SEM, TEM, and IR tests were carried out to analyze the effect of sintering temperature on the phase, crystallinity, and particle size of the samples. The results showed that the pure Mg_2_MnO_4_ phase could be obtained at temperatures of 500–700 °C. All the samples had a porous-spongy structure; the particle size was 30–50 nm for the samples annealed at 500–600 °C and increased to 100–200 nm when the temperature was up to 700–800 °C. The coordination environments and valence states of metal ions in the samples were analyzed by XPS, and the inversion parameters were calculated. It was found that manganese existed in three valence states in the Mg_2_MnO_4_ samples, in which Mn^3+^ and Mn^4+^ were located in the octahedral sites, and Mn^2+^ was located in the tetrahedral sites. With the increase in the heat treatment temperature of the samples, the inversion parameters and degree of cation disorder decreased. The electrochemical properties of Mg_2_MnO_4_ powders were also studied. It could be found that Mg_2_MnO_4_ as the lithium anode material had good cycling stability. The sample annealed at 600 °C had an initial discharge capacity of 771.5 mAh/g and a capacity of 340 mAh/g after 100 cycles. Mg_2_MnO_4_ was a potential anode material for lithium-ion batteries, and it was meaningful to improve the performance by decreasing the particle size and the degree of inversion of the material.

## Figures and Tables

**Figure 1 nanomaterials-11-01122-f001:**
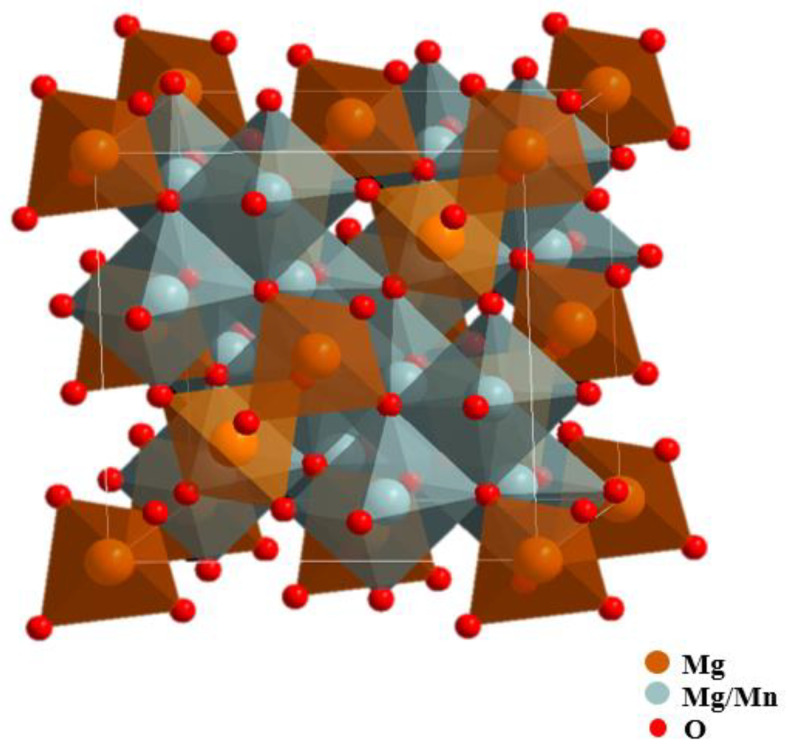
Theoretical structure of the cubic-phase Mg_2_MnO_4_ spinel.

**Figure 2 nanomaterials-11-01122-f002:**
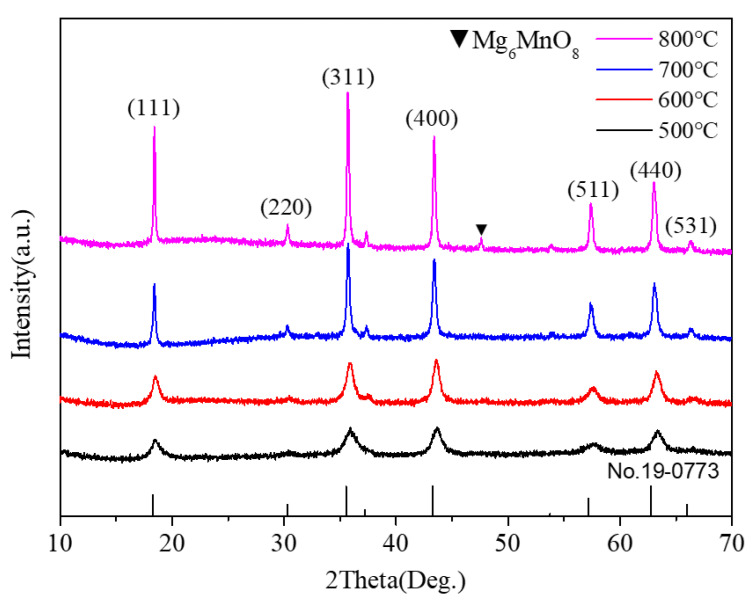
XRD pattern of Mg_2_MnO_4_ samples annealed at different temperatures.

**Figure 3 nanomaterials-11-01122-f003:**
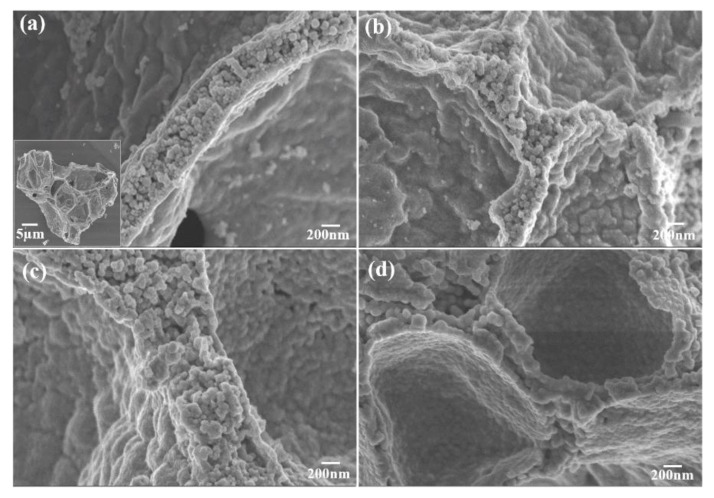
SEM images of Mg_2_MnO_4_ nanoparticles annealed at different temperatures: (**a**) 500 °C, (**b**) 600 °C, (**c**) 700 °C, (**d**) 800 °C.

**Figure 4 nanomaterials-11-01122-f004:**
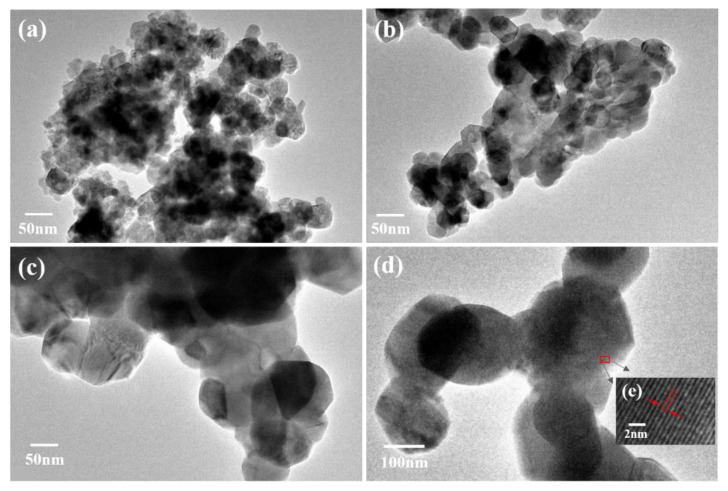
TEM images of Mg_2_MnO_4_ nanoparticles annealed at different temperatures: (**a**) 500 °C, (**b**) 600 °C, (**c**) 700 °C, (**d**) 800 °C, (**e**) HR-TEM image marked in (**d**).

**Figure 5 nanomaterials-11-01122-f005:**
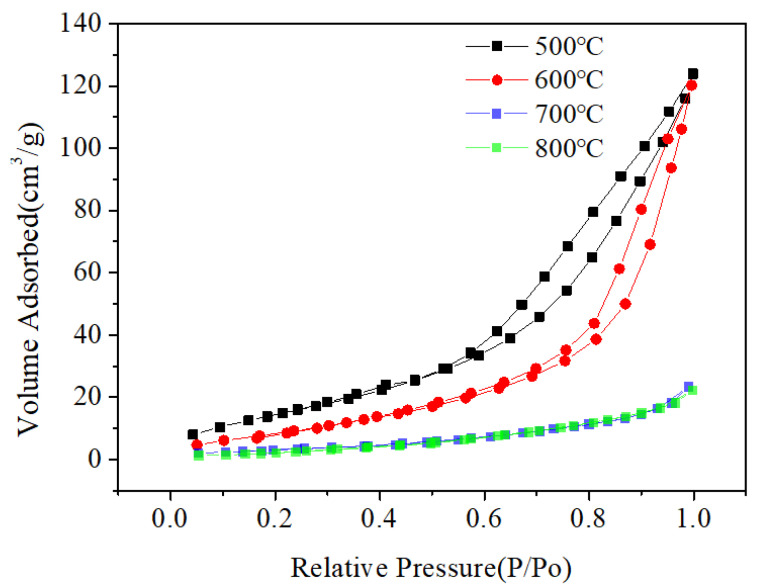
The N_2_ adsorption/desorption isotherms of Mg_2_MnO_4_ at different annealing temperatures.

**Figure 6 nanomaterials-11-01122-f006:**
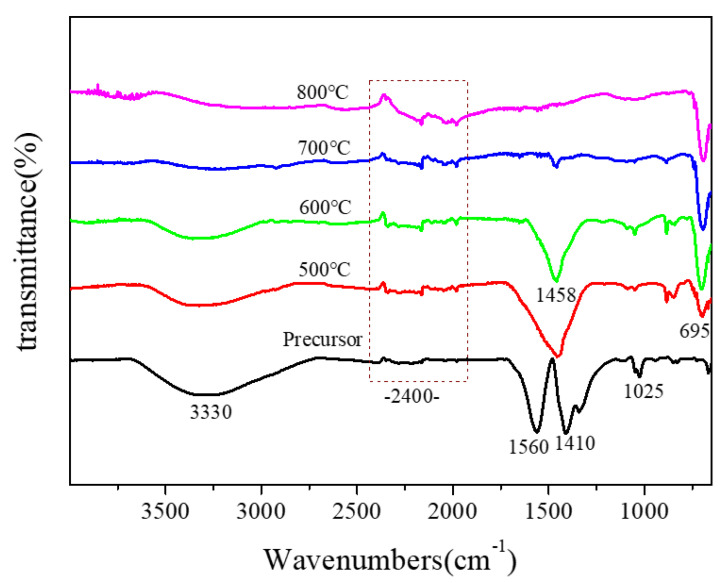
IR spectra of Mg_2_MnO_4_ samples annealed at different temperatures.

**Figure 7 nanomaterials-11-01122-f007:**
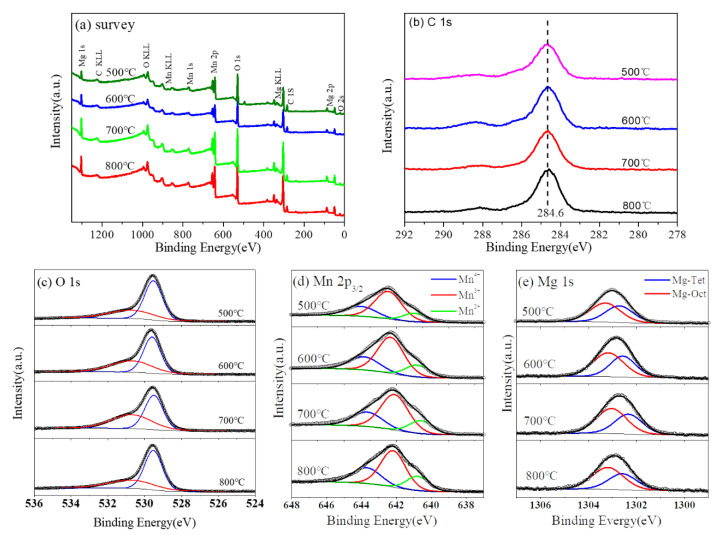
XPS spectra of Mg_2_MnO_4_ samples annealed at different temperatures: (**a**) survey, (**b**) C 1 s, (**c**) O 1s, (**d**) Mn 2p3/2, and (**e**) Mg 1 s.

**Figure 8 nanomaterials-11-01122-f008:**
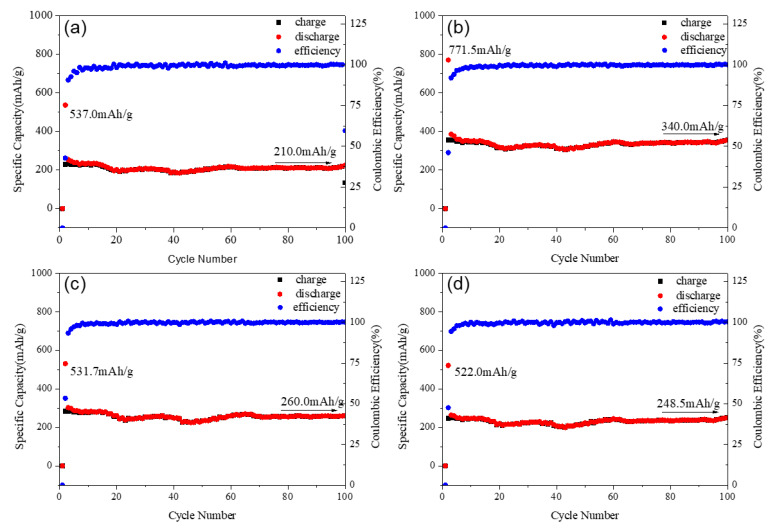
Specific capacity and cycle performance of Mg_2_MnO_4_ samples annealed at different temperatures: (**a**) 500 °C, (**b**) 600 °C, (**c**) 700 °C, (**d**) 800 °C.

**Figure 9 nanomaterials-11-01122-f009:**
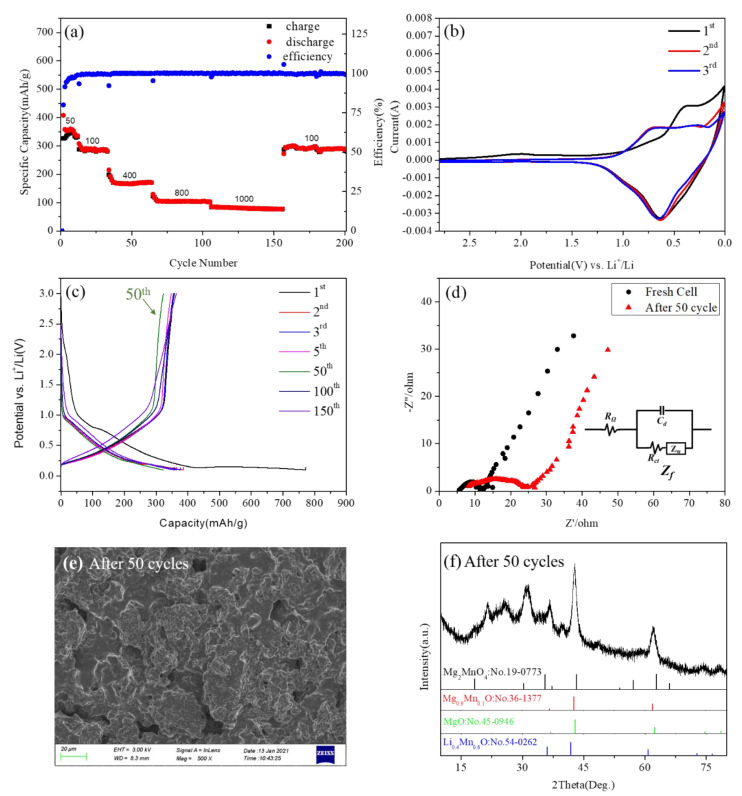
(**a**) The charge–discharge capacity of cycles for Mg_2_MnO_4_ at different current densities. (**b**) CV curves of Mg_2_MnO_4_ at a scan rate of 0.3mV/s. (**c**) The 1, 2, 3, 5, 50, 100, and 150 cycles of charge–discharge curves at a rate of 1C. (**d**) Nyquist plots of the electrode of Mg_2_MnO_4_ operating at the fresh cell and after 50 cycles. (**e**) The SEM image of the anode material after 50 cycles. (**f**) The XRD pattern of the anode material after 50 cycles.

**Table 1 nanomaterials-11-01122-t001:** XPS data for Mn 2p_3/2_ spectra of Mg_2_MnO_4_ nanoparticles annealed at different temperatures.

Temperature (°C)	Binding Energy (eV)	Fwhm (eV)	Area Percent (%)	Inversion Parameter
500	643.8	2.1	28.85	0.87
	642.3	1.9	58.17	
	640.8	1.6	12.98	
600	643.8	2.1	25.98	0.85
	642.3	1.8	58.67	
	640.8	1.5	15.35	
700	643.7	2.1	24.93	0.82
	642.1	1.9	57.11	
	640.6	1.6	17.96	
800	643.8	2.1	24.21	0.80
	642.3	1.8	55.32	
	640.8	1.6	20.47	

**Table 2 nanomaterials-11-01122-t002:** XPS data for Mg 1 s spectra of Mg_2_MnO_4_ nanoparticles annealed at different temperatures.

Temperature (°C)	Binding Energy (eV)	Fwhm (eV)	Area Percent (%)	Inversion Parameter
500	1303.3	1.6	55.71	0.89
	1302.7	1.4	44.29	
600	1303.2	1.6	57.68	0.85
	1302.6	1.3	42.32	
700	1303.0	1.5	58.15	0.84
	1302.3	1.4	41.85	
800	1303.2	1.6	60.41	0.79
	1302.6	1.4	39.59	

## Data Availability

The data presented in this study are available on request from the corresponding author.

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
