# Peer review of "Synthesis, Electronic Structure, and Electrochemical Properties of the Cubic Mg2MnO4 Spinel with Porous-Spongy Structure"

_nanomaterials, 2021, doi:10.3390/nano11051122_

Round 1

Reviewer 1 Report

In this contribution, Mg2MnO4 inverse-spinel has been prepared by sol-gel method, the resulting powders were characterized by different structural, microstructural and electrical techniques for potential application as lithium anode material. The following issues should be addressed before considering this paper for publication:

  1. The reagent manufacturer should be included in the experimental section.
  2. The structural characterization needs to be improved. The XRD data are not acquired with sufficient quality. It is recommendable to acquire the data with better resolution to visualize possible small diffraction peaks assigned to secondary phases.
  3. It is not clear how the cell parameters were determined: by indexation, Rietveld method, Le Bail, etc.. The error in the determination of the cell parameters could be included to estimate the quality of the fitting. It should be also useful to determine the crystal size at different annealing temperatures by using the Scherrer equation. These data could be compared with those obtained by SEM image.
  4. Regarding the SEM images (Fig. 3), these show a sponge-like morphology with large aggregates of particles, no suitable for electrochemical characterization. Have the authors studied the BET surface area with increasing annealing temperature?
  5. Please, revised the temperature value in Fig. 5.
  6. The authors deal that the samples are mixed spinels by XPS. However, XPS is a surface technique, not suitable to study the bulk composition of the materials. Surface contamination is not ruled out, since the sample contain water and carbon species, according to IR spectra. To confirm these results the O1s and C1s signals should also be analyzed. The local structure could be better studied by Raman spectroscopy.
  7. Please, check the assignation of the (440) crystal plane to lattice spacing 0.48 nm (page 4).
  8. The charge/discharge curves were taken after 150 cycles; however, the impedance spectra of fresh and 50 cycled sample are compared. It is recommendable to include the impedance spectra and SEM image for 150 cycles (Fig 9d).
  9. The authors deal on page 11 that the structure is decomposed; however, this is not confirmed by a structural analysis, such as XRD.
  10. It is not clear if the high frequency contribution of the impedance spectra is a depressed arc or is formed by different overlapped contributions. The discussion of the impedance spectra should be improved, for example, the contribution associated with the interception of the impedance spectra with the real axis is not commented. In addition, this contribution increases after cycling.
  11. The results need to be compared with those previously reported in the literature.
  12. The language should be carefully revised:

-The abstract should be written in the present tense instead of the past tense.

-Page 9: “are shown” instead of “were shown”

-Page 10 “the results indicate that”

-Page 10: “was restored… was restored..”

-Page 11: Fig. 9d instead of 9e

Reviewer 2 Report

In this manuscript, Mg2MnO4 nanoparticles with cubic spinel structure were synthesized by sol-gel method using polyvinyl alcohol as chelating agent. X-ray powder diffraction, infrared spectrum, scanning electron microscope and transmission electron microscope were used to characterize the crystalline phase and particle size of synthesized nanoparticles. The electronic structure and inversion degree of Mg2MnO4 spinel were studied by using X-ray photoelectron spectroscopy.

This manuscript is well written. The Introduction section is clear and concise. The experimental part is well presented. This manuscript can be publishable in Nanomaterials. However, two minor aspects should be considered:

1.- Acronym PVA is not defined in the Abstract.

2.- In the text around Figure 9 should be indicated that the Nyquist plots of the impedance show the imaginary part with minus sign versus the real part. The high frequency region is not a semicircle but a semicircular arc (it is not completed). Moreover, it should be pointed out that the Nyquist plots in Figure 9d are characteristic of electrochemical devices to energy storage and they can be appropriately interpreted by using a Randles circuit such as it is done in:

A.A. Moya, Journal of Power Sources, 397 (2018) 124-133.

Reviewer 3 Report

The manuscript “Synthesis, electronic structure and electrochemical properties of cubic Mg2MnO4 spinel with porous spongy structure” reports the preparation of pure Mg2MnO4 nanoparticles with cubic spinel structure using a new procedure based on sol-gel method and PolyVinyl Alcohol as chelating agent. The obtained structures were characterized: spongy porous structures made of particles of different sizes were observed, depending on the annealing temperatures chosen. The electrochemical performance  of  Mg2MnO4 as  lithium  anode  material  was  studied.  Results  showed  high  coulombic  efficiency  and  good  cycling stability, in particular for samples annealed  at  600oC. The effect of  calcination  temperature  on  the  charge-discharge  performance  of  the  samples  was  studied  and discussed. Perspectives to improve the performance include decreasing the particle size and the degree of inversion of the material.

The work is interesting, well done and generally well written. I recommend publication after minor corrections.

Here are suggestions for improvement:

1) Line 22 « Mg2MnO4samples  showed  higher  coulombic  efficiency  (above  98%)”: something is missing in this comparative sentence. Higher than what?

In several points of the discussion of results (electrochemical and performances studies), comparisons of the presented results to some state of the art materials and their performances are missing. Please add the appropriate references and numbers, for a better understanding of the reader.

2) Line 299: please correct the sentence “due to the destroy of the structure ». At this point, some more explanations are needed. What type of structure changes do you suspect?

This part ins not clear enough.

On one side, line 274 states that samples are stable until 150 cycles: “we can see that from the second cycle to 150 cycles, the sample exhibited almost no capacity attenuation and excellent cycle stability”.

On the other side (later in the paper), changes in structure are observed after 50 cycles using EIS, CV, SEM. For example, Line 301-302, “by comparing the SEM images after 50 cycles of charge/discharge (Fig. S5), the morphology of all samples had irreversible damage.” What type of damages? Please be more specific for a better understanding of the reader.

Round 2

Reviewer 1 Report

The authors have addressed most of the questions raised by the reviewers; however, some revisions are still necessary:

-The error in the determination of the cells parameters is not given. This is useful to  guarantee the quality of the fitting, since the cell parameters of the different samples is similar.

-The authors deal in the abstract “The results show that all the  Mg2MnO4 samples performance good cycle stability, and the coulombic efficiency in most cycles is near to 100%” However, the most relevant result is the decrease of the capacitance after 50 cycles due to phase composition..
